# Synergistic Effect of Bioactive Monoterpenes against the Mosquito, *Culex pipiens* (Diptera: Culicidae)

**DOI:** 10.3390/molecules27134182

**Published:** 2022-06-29

**Authors:** Amal Ramzi, Abdelhakim El Ouali Lalami, Saoussan Annemer, Yassine Ez zoubi, Amine Assouguem, Mikhlid H. Almutairi, Mohamed Kamel, Ilaria Peluso, Sezai Ercisli, Abdellah Farah

**Affiliations:** 1Laboratory of Applied Organic Chemistry, Faculty of Sciences and Technologies, Sidi Mohamed Ben Abdellah University, Route d’Imouzzer, Fez 2202, Morocco; ramzi.amal@usmba.ac.ma (A.R.); eloualilalami@yahoo.fr (A.E.O.L.); saoussan.annemer@usmba.ac.ma (S.A.); y.ezzoubi@uae.ac.ma (Y.E.z.); farah.abdellah1@gmail.com (A.F.); 2Higher Institute of Nursing Professions and Health Techniques, El Ghassani Hospital, Regional Health Directorate, Fez 30000, Morocco; 3Biotechnology, Environmental Technology and Valorization of Bio-Resources Team, Department of Biology, Faculty of Science and Techniques Al-Hoceima, Abdelmalek Essaadi University, Tetouan 2117, Morocco; 4Laboratory of Functional Ecology and Environment, Faculty of Sciences and Technologies, Sidi Mohamed Ben Abdellah University, Imouzzer Street, Fez 2202, Morocco; 5Zoology Department, College of Science, King Saud University, P.O. Box 2455, Riyadh 11451, Saudi Arabia; malmutari@ksu.edu.sa; 6Department of Medicine and Infectious Diseases, Faculty of Veterinary Medicine, Cairo University, Giza 12211, Egypt; m_salah@cu.edu.eg; 7Research Centre for Food and Nutrition, Council for Agricultural Research and Economics (CREA-AN), 00184 Rome, Italy; i.peluso@tiscali.it; 8Department of Horticulture, Faculty of Agriculture, Ataturk University, 25240 Erzurum, Turkey; sercisli@gmail.com

**Keywords:** *Culex pipiens*, insecticides, 1,8-cineole, α-pinene, binary mixture, fumigant

## Abstract

Mosquitoes represent one of the most important vectors and are responsible for the transmission of many arboviruses that affect human and animal health. The chemical method using synthetic insecticides disturbs the environmental system and promotes the appearance of resistant insect species. Therefore, this study investigated the insecticidal effect of some binary monoterpene combinations (1,8 cineole + α-pinene and carvone + R (+)-pulegone) using a mixture design approach. The fumigant toxicity was evaluated against *Culex pipiens* female adults using glass jars. The results show that the toxicity varies according to the proportions of each compound. Indeed, Mixture 1 (1,8-cineole + α-pinene) displayed a strong toxic effect (51.00 ± 0.86% after 24 h and 100.00 ± 0.70% after 48 h) when the pure compounds were tested at 0.25/0.75 proportions of 1,8-cineole and α-pinene, respectively. Nevertheless, the equal proportion (0.5/0.5) of carvone and R (+)-pulegone in Mixture 2 exhibited a toxic effect of 54.35 ± 0.75% after 24 h and 89.96 ± 0.14% after 48 h, respectively. For Mixture 1, the maximum area of mortality that the proposed model indicated was obtained between 0/1 and 0.25/0.75, while the maximum area of mortality in the case of Mixture 2 was obtained between 0.25/0.75 and 0.75/0.25. Moreover, the maximum possible values of mortality that could be achieved by the validated model were found to be 51.44% (after 24 h) and 100.24% (after 48 h) for Mixture 1 and 54.67% (after 24 h) and 89.99% (after 48 h) for Mixture 2. It can be said that all purev molecules tested through the binary mixtures acted together, which enhanced the insecticide’s effectiveness. These findings are very promising, as the chemical insecticide (deltamethrin) killed only 19.29 ± 0.01% and 34.05 ± 1.01% of the female adults after 24 h and 48 h, respectively. Thus, the findings of our research could help with the development of botanical insecticides that might contribute to management programs for controlling vectors of important diseases.

## 1. Introduction

Mosquitoes (Diptera: Culicidae) represent one of the most important vectors in the Arthropoda group and are well known due to the high risk they pose to human and animal health [1]. *Culex*, *Aedes*, and *Anopheles* are responsible for the transmission of a large number of dangerous parasites and pathogens that constitute the principal cause of the emergence of infectious diseases such as dengue fever, malaria, yellow fever, Zika virus, chikungunya virus, Japanese encephalitis, filariasis, Rift Valley fever, and West Nile virus, which cause an immense number of deaths each year [2,3,4,5,6,7,8]. In fact, the spread of arboviruses is apparently connected with many factors, such as climate change, urbanization, and the evolution of anthropic activities such as travel [9]. Recently, outbreaks caused by transmitted viruses have attracted the attention of the whole world and created a serious public health problem.

Despite the progress in vaccine development, no efficient human vaccine is available for certain mosquito-borne diseases such as the West Nile virus where the genus *Culex*, and in particular the mosquito *Culex pipiens*, is suspected to be responsible for the transmission in some countries, including Morocco [10,11,12]. It is the principal mosquito species that appears to be infected with West Nile virus in the field [13]. Thus, the best way to protect against vectors is to avoid mosquito bites [4,14], and efficient vector management strategies remain crucial tools to control and prevent the spread of mosquito-borne diseases [15]. Actually, bio-control occupies an important place in vector management programs, especially with regard to problems associated with the use of chemical insecticides, such as environmental pollution, the risk to human health, and the appearance of resistant mosquito species. Over the last few years, researchers have focused on the investigation of botanical insecticides from plant extracts and evaluated the insecticidal potential of several chemical constituents isolated from plant-derived essential oils in an attempt to determine the most bioactive chemical compounds. Indeed, essential oils constitute a mixture of molecules while monoterpenes, sesquiterpenes, and phenylpropenes represent the main classes of aromatic plant-based essential oils (EOs) [16].

Monoterpenes have attracted the attention of scientists because of their broad spectrum of biological properties, and their usage in insect control has been widely investigated [16,17]. These secondary metabolites have been shown to have efficacy against both the larvae and adults of many mosquito species [18,19,20,21,22]. Therefore, this study aimed to evaluate some monoterpenes in a binary combination using a mixture design method against *Culex pipiens* adult females. To the best of our knowledge, studies have not yet been conducted in the context of monoterpene mixtures using the above-stated approach.

## 2. Materials and Methods

### 2.1. Materials

The pure compounds 1,8-cineole (99%), α-pinene (98%), Carvone (98%), and R (+)-pulegone (99%) (Figure 1) were purchased from Sigma-Aldrich (Darmstadt, Germany). Deltamethrin, obtained from the Scientific Fertilizer Co. (P) Ltd. (Coimbatore, India), was used as a positive control at the diagnostic dose (0.05%) suggested by the World Health Organization (Geneva, Switzerland) for adult mosquitoes.

### 2.2. Preparation of the Binary Mixture

Based on the LC_50_ values (Table 1) obtained for the individual compounds in a previous study (Ramzi et al., 2022) [23], seven binary combinations for 1,8 cineole + α-pinene and carvone+ R (+)-pulegone were prepared using a binary mixture design as described in Table 2. For each experiment, the two compounds were combined using the various proportions that refer to volume.

In this study, a simplex-lattice design for two compounds was selected. This mixture design was established by seven experiments (lattice degree, 3) with 2 pure compounds (Experiments 1 and 5), the mixture 0.5–0.5 (Experiment 7), axial points (Experiments 2 and 3), and two mixtures (0.6–0.3) of the two constituents. The suggested mathematical model is a quadratic model of Scheffe as indicated in the Equation (1) below:(1)$$Y=b_{1}x_{1}+b_{2}x_{2}+b_{12}x_{1}x_{2}+\varepsilon$$
where ***Y*** is the response (Mortality %), ***b*_1_**, ***b*_2_**, and ***b*_12_** are the coefficients of the linear terms and the quadratic term, respectively, ***X_i_*** is the proportion of each compound, and ***ε*** is an error term. However, one restriction on the selected combinations is that the amount of each pure compound should vary from 0 to 1, and the total amount of all components used in the mixture should be equal to 1 as indicated in Equation (2).
(2)$$\overset{i=n}{\underset{i=1}{\sum }}X_{i}=1$$

### 2.3. Mosquitoes

Larvae of *Culex pipiens* were gathered from a beading site named Oued El Mehraz (Fez—Northeast of Morocco) using a rectangular plastic tray that was placed in an inclined position (45°) with respect to the water surface. In the laboratory, the collected larvae were maintained in cages with dimensions of 24 × 24 × 24 cm^3^ under an average temperature of 22.6 ± 2 °C and a relative humidity of 70 ± 5%. The emerging adults were then fed on a 10% sucrose solution. The morphological identification of *Culex pipiens* adults was performed using the Moroccan identification key of Culicidae and the mosquitoes of Mediterranean Africa identification software [24,25].

### 2.4. Fumigant Toxicity

The fumigant test was performed using glass jars (1 L) and by following the method adopted by Zahran et al. (2017) and Badawy et al. (2017) [26,27] with slight modifications. The toxicity was evaluated on *Culex pipiens* female adults (2–3 days). A total of 5 µL of each mixture (previously prepared as mentioned in the “Preparation of Binary Mixture” section) was applied on Whatman N°1 filter papers with a 7 cm diameter. These papers had already been placed on the lower surface of the glass jar covers. The chemical insecticide Deltamethrin was used as a positive control. Dimethylsulfoxyd (DMSO) was used as a negative control. Three replicates were performed for each binary combination and for the controls with 20 adult females per test. The mortality rates were recorded after 24 h and 48 h of exposure to the mixtures and the insecticide.

### 2.5. Statistical Treatment

Mortalities were calculated using Abbott’s Equation (3). The DESIGN EXPERT software version 12 (Stat-Ease society, (Minneapolis, MN, USA) version 12) was used for the mixture design treatment. The obtained results were analyzed by ANOVA to test the significance of the postulated models. All tests were performed at a 95% significance level.
(3)$$\%\;\mathrm{Mortality}\;\mathrm{Corrected}=\left[\frac{\%\;\mathrm{Mortality}\;\mathrm{Observed}-\%\;\mathrm{Mortality}\;\mathrm{Control}}{100-\%\;\mathrm{Mortality}\;\mathrm{Control}}\right]\times 100.$$

## 3. Results

### 3.1. Acute Toxicity of the Tested Binary Combinations

Table 3 presents the results obtained for the binary mixtures 1,8-cineole + α-pinene and carvone + R (+)-pulegone. It can be seen that all of the selected monoterpenes had insecticidal efficacy against *C. pipiens* adults at all combinations tested. For the binary combination of 1,8-cineole + α-pinene, the results reveal that Experiments 1, 3, 4, and 5 were more effective than the other tests after 24 h, while the mortalities were around 50%. Experiments 2, 3, and 4 gave high mortality rates after 48 h with toxicities between 90.00 ± 0.70% and 100.00 ± 0.70%. For the carvone and R (+)-pulegone mixture, it appears that both compounds work together, especially when we look at Experiment 7 (0.5/0.5) as the mortalities increased (they were found to be 54.35 ± 0.75% after 24 h and 89.96 ± 0.14% after 48 h). On the other hand, the synthetic insecticide displayed a low mortality rate (19.29 ± 0.01% after 24 h and 34.05 ± 1.01% after 48 h) in comparison with the two monoterpene combinations.

### 3.2. The Mixture Design Treatment

#### 3.2.1. Statistical Validation of the Selected Model

The analysis of variance (Table 4) showed that there is a significant main effect of regression for all the selected mixtures regardless of the time of exposure as the probabilities of the risk significance *p*-value are less than 0.05 at the 95% confidence level. For the two mixtures, the coefficients of determination are equal to 99%; this value shows the good fit of the model, which means that there is good agreement between the experimental and predicted values of the adapted model. Figure 2 confirms these results since the curves of the predicted values as a function of the actual values present a line shape.

#### 3.2.2. Compound Effects

The effects of the two binary combinations as well as the *t*-Student statistical values and the observed probability (*p*-value) are presented in Table 5. It seems that all of the model coefficients are statistically significant as their *p*-values are less than 5%. Therefore, all of these coefficients must be kept from the proposed model.

A simplex-lattice design was adopted to model the mortality as a function of compound variables and is represented by the following equations:-For Mixture 1 at 24 h:
Y1 = 49.10 X1 + 49.93 X2 − 40.51 X1X2 + ε(4)-For Mixture 1 at 48 h:
Y2 = 80.69 X1 + 84.44 X2 − 27.22 X1X2 + ε(5)-For Mixture 2 at 24 h:
Y3 = 51.62 X1 + 50.87 X2 + 13.14 X1X2 + ε(6)-For Mixture 2 at 48 h:
Y4 = 84.20 X1 + 80.45 X2 + 31.67 X1X2 + ε(7)

#### 3.2.3. The Maximum Response Zones (Isoresponses)

Figure 3 demonstrates the variation in mortality percentages according to the amount of the selected compound. From Figure 3, we can consider the different activities relative to the different proportions of the two compounds due to the mixture profile. The profile of the mixture shows that desirable insecticidal activity (>50%) is possible when the line (mortality) increased. For Mixture 1 (1,8 cineole + α-pinene), the maximum mortality rate was obtained between 0/1 and 0.25/0.75 at 24 and 48 h. The optimum mortality rate in the case of Mixture 2 (carvone+ R (+)-pulegone) ranged between 0.25/0.75 and 0.75/0.25 after 24 h and 48 h of treatment.

#### 3.2.4. Desirability Function

In order to reach the maximum possible value for the studied response (Mortality %), the desirability function was applied. Figure 4 and Figure 5 represent the desirability plot that was achieved by the validated model for the mortality of both mixtures at 24 h and 48 h, respectively.

Figure 4a shows that the maximum value of the mortality percentage obtained by Mixture 1 at 24 h is equal to 51.44% when the proportions of 1,8-cineole and α-pinene are equal to 0.27 and 0.73, respectively. Figure 4b illustrates that the mortality reaches its maximum value of 100.24% for Mixture 1 at 48 h when the proportions of 1,8-cineole and α-pinene attain values of 0.25 and 0.75, respectively. Figure 5a indicates that the mortality reaches its maximum value of 54.64% for Mixture 2 at 24 h when the proportions of carvone and R (+)-pulegone are equal to 0.48 and 0.52, respectively. Figure 5b presents the maximum value of the mortality rate achieved by Mixture 2 at 48 h, which is equal to 89.99% when the proportions of carvone and R (+)-pulegone reach values of 0.55 and 0.45, respectively.

## 4. Discussion

In our study, the two binary mixtures displayed remarkable toxicity towards *C. pipiens* adults, and the exerted effect varied as a function of the proportion of every single compound. As the results show, the combination of 1,8-cineole + α-pinene was found to be more effective than the mixture of carvone + R (+)-pulegone. However, in general, the two binary mixtures showed significantly higher insecticidal potential against *C. pipiens* female adults compared with that observed with the synthetic insecticides.

Nowadays, bio-control with natural plant products plays an interesting role in mosquito management programs. These botanical insecticides represent a suitable alternative to the chemical insecticides; they naturally contain many bioactive molecules that have been proven to have insecticidal effectiveness, and, on the other hand, they importantly pose only a minimal risk to human and animal health and the environment. For these reasons, they have attracted the interest of scientists all over the world. Indeed, many studies have reported on the efficacy of plant extracts, including essential oils and their derived compounds. In the last few years, researchers have focused on the bioactive molecules of EOs and have isolated and evaluated the toxicity of different compounds, such as monoterpenes [1,21,28].

In fact, the chemical profile of many aromatic plants shows that monoterpenes (hydrocarbon or oxygenated monoterpenes) have the highest abundance, while the major constituents of EOs belong to this chemical class. Thus, monoterpenes have been broadly tested as insecticidal agents for insect control [15]. They are well known and well documented to be active fumigants, repellents, and adulticides towards many insect species [29].

The four monoterpenes evaluated through the binary mixtures in our study constitute the major constituents in many EOs, such as *Rosmarinus officinalis*, *Mentha pulegium*, and *Lavandula* [30,31,32]. Indeed, 1,8-cineol (a monoterpene ether) has been reported to be toxic to many insect species, including *C. pipiens* [21]. Similarly, α-pinene (a monoterpene hydrocarbon) was found to be a toxic fumigant compound [18,33]. The two compounds used in the second mixture, carvone and R (+)-pulegone, belong to the monoterpene oxygenated fraction (ketones) and have also been proven to have insecticidal potential on *C. pipiens* larvae and adults [21].

The synergistic effects of plant-derived EO compounds on female adult mosquitoes have not been widely reported. However, these effects have been investigated against mosquito larvae and other insect species. Badawy et al., 2016 [34] evaluated the fumigant effect of monoterpene formulations using different classes, including those belonging to the monoterpene ketones. They concluded that the mixture prepared with camphor, menthone, carvone, and fenchone on a paper disc was the most effective against *C. pipiens* adults. These results were in agreement with those of Ma et al., 2014 [33], who investigated the combined effects of certain EO compounds through fumigation tests on *C. pipiens* pallens adults. Ma and his colleagues found that a mixture of carvacrol and thymol exhibited strong fumigant toxicity and showed a synergistic effect with a co-toxicity factor of 83.1.

Other monoterpenes (thymol, carvacrol, linalool, and methyl cinnamate) alone and in combination have also been evaluated against *C. pipiens* larvae and larvae of other mosquito species [22,35]. Monoterpenes have also been evaluated against other insect species [36,37]. Moreover, Sarma et al., 2019 [38] tested the synergistic and antagonistic effects of many essential-oil-based terpene compounds against larvae and adult mosquitoes from the Culicidae family (*Aedes aegypti*), where the combination of carvone and limonene exhibited the best adulticidal toxicity and a synergistic effect (X^2^ = 17.72). Against other insect genera, Choi et al., 2006 [18] revealed that the fumigant toxicity of a combination of α- and β-pinene against mushroom fly adults was better than the pinene compounds alone. By reducing the concentrations tested and enhancing the insecticidal effect against the target organism, the synergistic mixtures yielded great results [37]. All of these studies demonstrated the toxic effect of EO-derived compound mixtures on several insects from different Arthropoda families. The insecticidal activity could be attributed to the interaction between the various constituents of the mixture [39].

The toxic action modes of these bioactive compounds remain poorly understood. Nevertheless, researchers have reported some variability in the action mechanisms of monoterpenes depending on the insect target sites [18]. Ryan and Byrne (1988) [40] investigated the relationship between insecticidal effects and monoterpene toxicity. They indicated that these constituents act on the cholinergic system through the inhibition of acetylcholinesterase (AChE). This finding was in accordance with the results of Enan (2001) [41], who reported that EO components exert a neurotoxic action and found that the toxicity was related to the octopaminergic nervous system of insects, as the most important symptoms were hyperactivity followed by hyperexcitation leading to rapid knockdown and immobilization. Moreover, Burčul et al. (2020) [42] reported in their recent review that many studies have proved the effectiveness of EO compounds, such as α-pinene, δ-3-carene, 1,8-cineole, carvacrol, thymohydroquinone, α- and β-asarone, and anethole, which showed promising cholinesterase inhibitory activity. Lee et al. (2001) [43] suggested that the toxicity exerted by terpenic compounds is not necessarily correlated with their ability to inhibit the action of AChE and that monoterpene compounds may be responsible for inhibiting cytochrome-P450-dependent monooxygenases.

These findings indicate that plants and their secondary metabolites, such as EOs, are promising biological alternatives given the impact of chemical control programs on humans and the environment. Indeed, EOs and their components constitute safer and more eco-friendly mosquito control tools. Many bioactive molecules are selective and less toxic to humans, animals, and the environment [3,44,45].

Currently, mosquito bio-control is one of the five pivotal tactics of Integrated Mosquito Management (IMM), which is considered to be the best way to control mosquitoes and decrease the transmission of arboviruses. This mosquito control program may adequately prevent mosquito-borne diseases or reduce their transmission within a larger eco-evolutionary context [46,47,48,49]. In fact, managing the spread of mosquitoes can reduce the effects of mosquito-borne disease epidemics. Thus, an appropriate and reasonable mosquito management approach could help to achieve the desired goal.

Therefore, our results serve as preliminary data for the development of new biological control agents that could be incorporated into mosquito control strategies, especially when we consider the emergence of species resistant to synthetic insecticides [50,51]. Overall, scientific investigations have an important role to play as they provide an evidence base that could help with management actions.

## 5. Conclusions

This study evaluated the insecticidal effect of monoterpenes in binary combinations on adult female mosquitoes of great public health importance, *C. pipiens*. Based on the obtained results, the bioactive molecule formulations could be used in vector programs as an eco-friendly, biological, and effective alternative for the development of new botanical insecticides that could be used in the impregnation of mosquito nets, to make mosquito traps for adults, or to develop botanical insecticides. Nevertheless, further studies should be carried out on the acetylcholinesterase and/or butyrylcholinesterase inhibitory potential as well as the biosafety of these natural substances. We also think that this research should be extended to the evaluation of the insecticidal potential of other EO bioactive molecules (monoterpenes and sesquiterpenes) as well as testing all possible combinations.

## Figures and Tables

**Figure 1 molecules-27-04182-f001:**
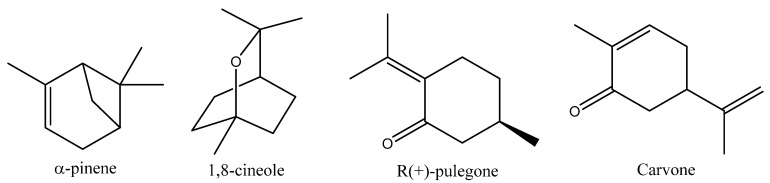
The molecular structures of the four monoterpenes as determined by ChemBioDraw software (version 16.0).

**Figure 2 molecules-27-04182-f002:**
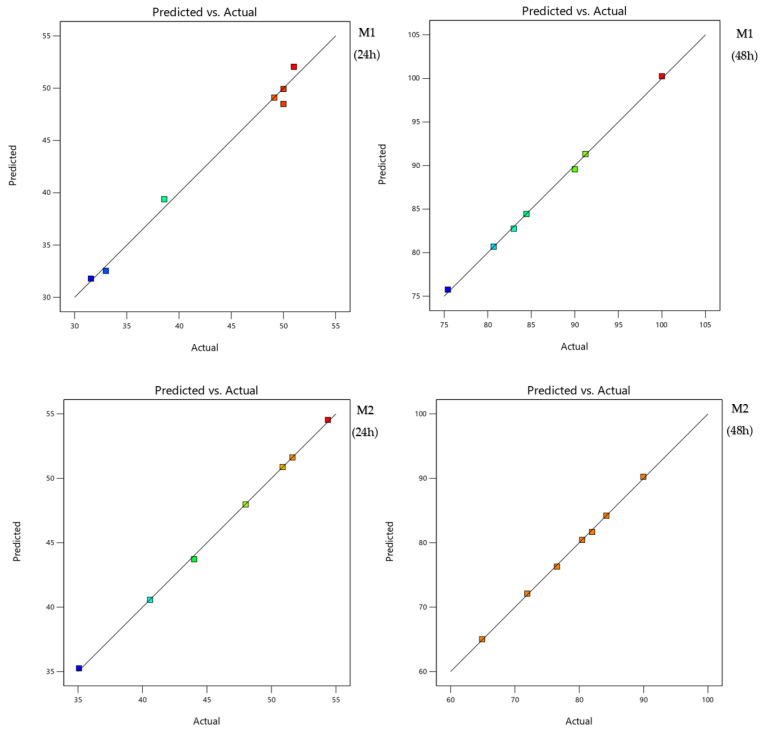
Curves of the predicted values according to the actual ones for both binary mixtures.

**Figure 3 molecules-27-04182-f003:**
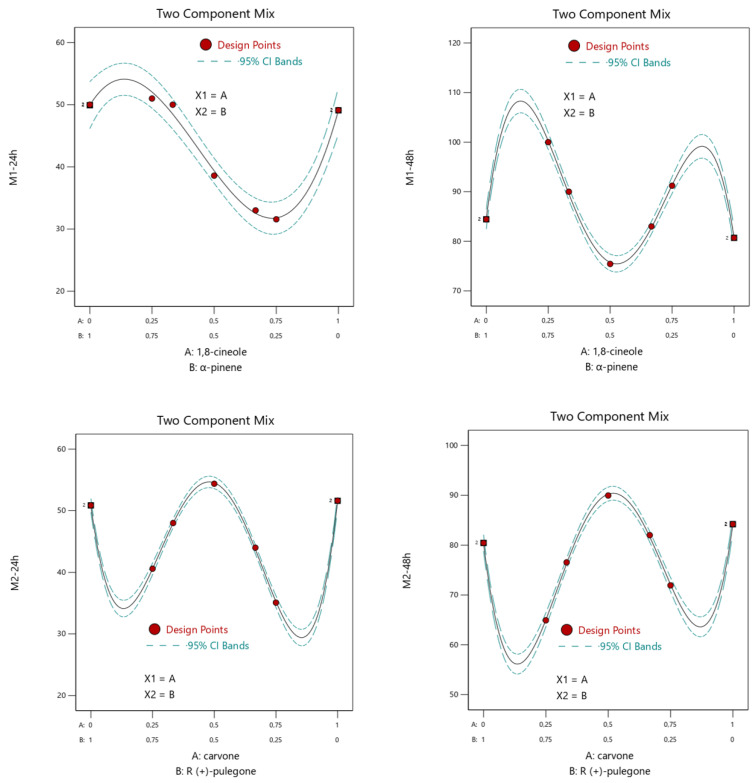
Binary combination graph showing the area of maximization of the response studied as a function of the compound proportion. M1, 1,8-cineole + α-pinene; M2, carvone + R (+)-pulegone.

**Figure 4 molecules-27-04182-f004:**
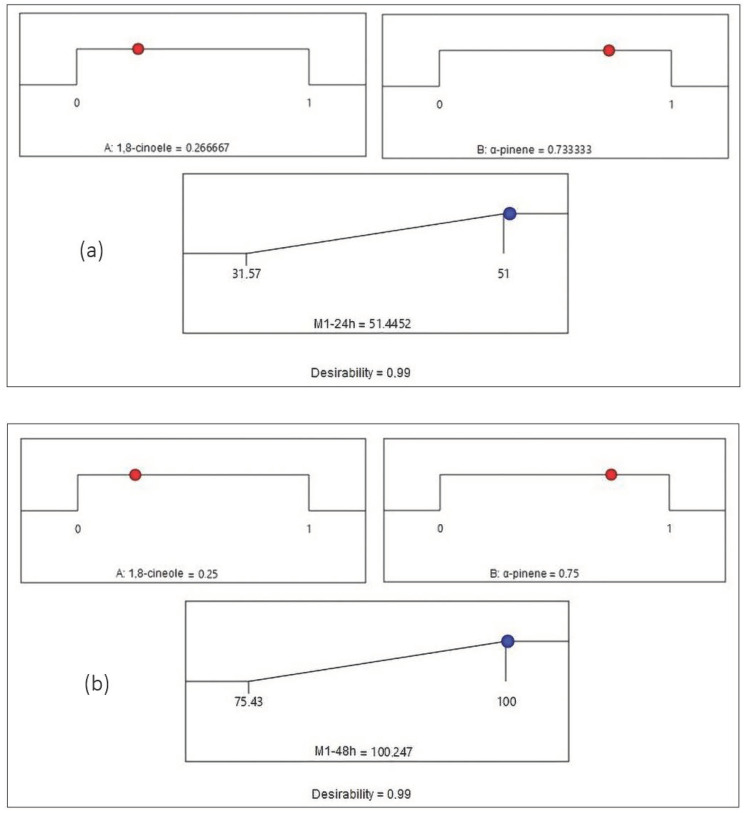
Desirability profile of optimal conditions to maximize the mortality rate for Mixture 1 (**a**) at 24 h and (**b**) at 48 h. (A) 1,8-cineole; (B) α-pinene.

**Figure 5 molecules-27-04182-f005:**
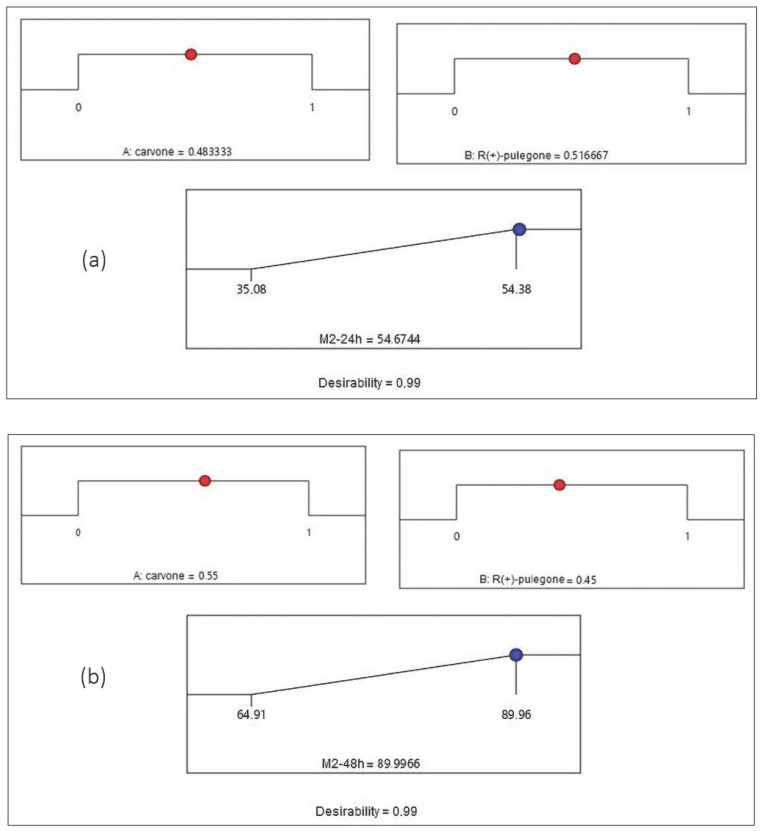
Desirability profile of optimal conditions to maximize the mortality rate for Mixture 2 (**a**) at 24 h and (**b**) at 48 h. (A) carvone; (B) R (+)-pulegone.

**Table 1 molecules-27-04182-t001:** Lethal concentration (LC_50_) values for the individual monoterpenes.

Compound	LC_50_ ^a^ (µL/L Air) (95% Confidence Intervals)	Slope ^b^	Intercept ^c^	R^2^	*p*-Value ^d^
**1,8-Cineole**	5395.65 (5282.45–5486.55)	0.362 ± 0.01	3.649 ± 0.03	0.98	0.000
**α-Pinene**	1294.64 (1230–1345.14)	0.642 ± 0.06	3.002 ± 0.16	0.92	0.000
**Carvone**	1713.36 (1703.33–1723.36)	0.774 ± 0.04	2.497 ± 0.12	0.96	0.000
**R (+)-Pulegone**	5395.58 (5295.58–5494.68)	0.515 ± 0.01	3.078 ± 0.02	0.99	0.000

^a^: Concentration that kills 50% of adult mosquitoes. ^b^: Slope of the regression line ± SE. ^c^: Intercept of the regression line ± SE. ^d^: Significant effect at a *p* value < 0.05.

**Table 2 molecules-27-04182-t002:** The seven experiments selected for each binary mixture.

Mixture	Components	
	A	B
1	0	1
2	0.75	0.25
3	0.25	0.75
4	0.33	0.67
5	1	0
6	0.67	0.33
7	0.5	0.5

**Table 3 molecules-27-04182-t003:** Toxicity of the two mixtures against *C. pipiens* female adults.

Binary Mixture (A + B) (LC_50_ + LC_50_)	Mortality % ± SE ^a^					
	1,8-Cineole + α-Pinene		Carvone + R (+)-Pulegone		Deltamethrin	
	(24 h)	(48 h)	(24 h)	(48 h)	(24 h)	(48 h)
1	50.01 ± 1.18	84.45 ± 0.11	50.87 ± 0.81	80.45 ± 0.10	19.29 ± 0.01	34.05 ± 1.01
2	31.57 ± 0.18	91.22 ± 0.11	35.08 ± 0.60	71.92 ± 0.70		
3	51.00 ± 0.86	100.00 ± 0.70	40.59 ± 0.41	64.91 ± 0.75		
4	50.00 ± 0.84	90.00 ± 0.70	48.00 ± 0.63	76.54 ± 0.67		
5	49.12 ± 1.01	80.69 ± 0.10	51.63 ± 0.85	84.21 ± 0.11		
6	33.00 ± 0.16	83.00 ± 0.10	44.00 ± 0.71	82.00 ± 0.10		
7	38.59 ± 0.47	75.43 ± 0.51	54.35 ± 0.75	89.96 ± 0.14		

^a^: Mortality rates are presented as means ± SE (*n* = 3).

**Table 4 molecules-27-04182-t004:** Analysis of variance for the adjusted model for all selected mixtures.

Source of Variance	Mixture	Exposure Time (h)	SS	DF	MS	F-Value	*p*-Value
Model	M1	24	444.65	3	148.22	103.76	0.001
		48	389.04	4	97.26	466.30	0.002
	M2	24	281.09	4	70.27	1062.82	0.000
		48	411.49	4	102.87	703.49	0.001
Residual	M1	24	4.29	3	1.43		
		48	0.417	2	0.20		
	M2	24	0.1322	2	0.0661		
		48	0.2925	2	0.1462		
Total	M1	24	468.94	6			
		48	389.46	6			
	M2	24	281.22	6			
		48	411.78	6			
R^2^	M1	24	0.99				
		48	0.99				
	M2	24	0.99				
		48	0.99				

M1, Mixture 1 (1,8-cineole + α-pinene); M2, Mixture 2 (carvone+ R (+)-pulegone); SS: sum of squares; DF, degrees of freedom; MS, mean square; R^2^, coefficient of determination.

**Table 5 molecules-27-04182-t005:** Estimated regression coefficients for the incomplete cube regression model.

	Time (h)	Coefficient	Estimation ± SE	*t*-Student	*p*-Value
1,8-cineole	24	b1	49.10 ± 1.20	41.42	<0.0001 *
	48	b1	80.69 ± 0.45	42.59	<0.001 *
α-pinene	24	b2	49.93 ± 1.20	42.11	<0.0001 *
	48	b2	84.44 ± 0.45	40.94	<0.001 *
1,8-cineole + α-pinene	24	b12	−40.51 ± 4.58	−18.85	0.0030 *
	48	b12	−27.22 ± 2.01	−13.53	0.0050 *
Carvone	24	b1	51.62 ± 0.26	−59.07	<0.0001 *
	48	b1	84.20 ± 0.38	−49.53	<0.0001 *
R (+)-pulegone	24	b2	50.87 ± 0.26	48.11	<0.0001 *
	48	b2	80.45 ± 0.38	52.53	<0.0001 *
Carvone+ R (+)-pulegone	24	b12	13.14 ± 1.13	11.60	0.0070 *
	48	b12	31.67 ± 1.68	18.80	0.0030 *

*: Statistically significant at a *p*-value < 0.05.

## Data Availability

All related data are contained within the manuscript.

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
