# Peer review of "Synergistic Effect of Bioactive Monoterpenes against the Mosquito, Culex pipiens (Diptera: Culicidae)"

_molecules, 2022, doi:10.3390/molecules27134182_

Round 1

Reviewer 1 Report

The manuscript is a resubmission of a paper that I was asked to review twice before. In my review of the resubmission, I noted that a core set of data (Table 3) had been changed, but that none of the other data tables or figures, some of which rely on the data in Table 3, had been revised. The current submission seems to have overlooked or ignored my comments submitted May 4. There has been no improvement to the manuscript since the last time I reviewed it and my recommendation remains “reject.”  

Author Response

The authors would like to express their gratitude to the reviewers for the revision of our manuscript entitled: “ Synergistic effect of bioactive monoterpenes against the mosquito, Culex pipiens (Diptera: Culicidae)”. We sincerely appreciate all valuable comments and suggestions.

Our responses to the Reviewers’ comments are described below. Appropriated changes, suggested by the Reviewers have been introduced to the manuscript (highlighted within the document in different color).

Reviewer 1

Comment:

The manuscript is a resubmission of a paper that I was asked to review twice before. In my review of the resubmission, I noted that a core set of data (Table 3) had been changed, but that none of the other data tables or figures, some of which rely on the data in Table 3, had been revised. The current submission seems to have overlooked or ignored my comments submitted May 4. There has been no improvement to the manuscript since the last time I reviewed it and my recommendation remains “reject.”

Response:

Dear reviewer, thank you very much for the revision of our resubmitted paper. Indeed, the authors didn’t ignore your past comments; we previously indicated in our responses of comments for the first submission that there were some inconsistences, especially in table 3, it was actually the wrong version and we apologized for that. After the recommendation and the remarks of the reviewer we corrected all the necessary data in table 3 and also in the abstract (lines 31-42) and the results part (lines 147-154) (highlighted in the paper with green color). The figures have been also revised; we remade all figures as recommended by a reviewer in the first submission and we also indicated some details in figures 3 and 4 (About legends: Design points, 95% CI bands…and desirability). Moreover, we noted that the justification for the trend lines in figure 3 has been added to the results section, the maximum response zones (Isoresponses) part, lines 194-200. The adopted mathematical model for two mixtures has been also added in the manuscript in the result section, statistical validation of the selected model part, lines 182-191. All corrections have been highlighted in the document in Green color. Thus, the authors have carefully revised the paper and corrected the inconsistency noticed by the reviewer.  We really hope that our responses will suit the reviewer’s expectations.

Reviewer 2 Report

Amal Ramzi and co-workers presented the results of a scientific study conducted on Culex pipiens mosquitoes in a paper entitled Evaluation of the synergistic effect of pure molecules on adults of Culex pipiens (Diptera: Culicidae). Namely, under controlled conditions, they were exposed to the action of combined different monoterpenes in different proportions. The paper's topic is current, the problem of vector diseases is ubiquitous, and the resistance of vectors to existing insecticides. Therefore, there is a need to find new means to suppress the vector.

I think that the work is well-conceived and that the work results are presented appropriately.

It is necessary to correct:

  1. Line 2: I think that the paper's title is redundant and that it should be modified "pure molecules" is too broad a term.

1.       Line 64: I think it would be better to say "principal" instead of "only"

„Mosquitoes of the genus Culex are generally considered the principal vectors of WNV, in particular Cx. pipiens.“ https://www.who.int/news-room/fact-sheets/detail/west-nile-virus

2.       Line 88: Deltamethrin – the manufacturer must be specified

The literature is not listed according to the journal's instructions, which is necessary to harmonize.

Author Response

The authors would like to express their gratitude to the reviewers for the revision of our manuscript entitled: “ Synergistic effect of bioactive monoterpenes against the mosquito, Culex pipiens (Diptera: Culicidae)”. We sincerely appreciate all valuable comments and suggestions.

Our responses to the Reviewers’ comments are described below. Appropriated changes, suggested by the Reviewers have been introduced to the manuscript (highlighted within the document in different color).

Reviewer 2

Comment 1: 

Line 2: I think that the paper's title is redundant and that it should be modified "pure molecules" is too broad a term.

Response 1:

We corrected the paper's title, and we also replaced the term “pure molecules”

“The corrections were highlighted in blue color”

Comment 2: 

Line 64: I think it would be better to say "principal" instead of "only"

Response 2:

We replaced, in the line 64, the word “only” with "principal".

Comment 3: 

„Mosquitoes of the genus Culex are generally considered the principal vectors of WNV, in particular Cx. pipiens.“ https://www.who.int/news-room/fact-sheets/detail/west-nile-virus

Response 3:

Thank you for this relevant information, we introduced it in our manuscript “Introduction part, line 63”

Comment 4: 

Line 88: Deltamethrin – the manufacturer must be specified

 Response 4:

We indicated the manufacturer “line 90”.

Comment 5: 

The literature is not listed according to the journal's instructions, which is necessary to harmonize.

Response 5:

We revised the references part.

Thank you again for your comments. We hope that our answers will suit your expectations.

Sincerely.

Round 2

Reviewer 1 Report

Too many significant figures are reported in Figure 4.

Author Response

Response to Reviewer

We would like to thank the reviewer for careful and thorough reading of this manuscript and for the thoughtful comments and constructive suggestions, which help to improve the quality of this manuscript. The responses to reviewer’ comment has been highlighted to the manuscript with a Grey color.

Comment 1:

Too many significant figures are reported in Figure 4.

Response 1:

Thank you for the nice suggestion and we completely agree with you. As suggested by the reviewer, figure 4 has been divided into two figures (4 and 5) for better visualization. Figures 4 and 5 represents the desirability plot of the two mixture M1 (1,8-cineole+ α-pinene) and M2 (Carvone+ R (+)-pulegone) to optimize the percent mortality with maximum desirability (0.99). The desirability function stays at value 0 as long as the response value is unsatisfactory and goes to 1 as soon as the response value is satisfactory. The desirability function is quite useful when it is necessary to find the best compromise among several responses. This function was proposed by Derringer and Suich (1980) and Jones (1990) and appears in several DOE software. We indicated more detail for Figures 4 and 5 (Lines 211-223).

Thank you again for your comments that permitted to increase the quality of our paper. We hope that our answers will suit the reviewer expectations.

Sincerely,

This manuscript is a resubmission of an earlier submission. The following is a list of the peer review reports and author responses from that submission.

Round 1

Reviewer 1 Report

Dear Authors,

this is an interesting article on the synergism of natural molecules against mosquitoes. Overall the work is well organized, analyzed and presented and the results well sustained. Thus I suggest acceptance for publishing only some language editing.

Author Response

Dear Editor,

The authors would like to express their gratitude to the reviewers for the revision of our manuscript entitled: “Evaluation of the synergistic effect of pure molecules on adults of Culex pipiens (Diptera: Culicidae)”. We sincerely appreciate all valuable comments and suggestions.

Our responses to the Reviewers’ comments are described below. Appropriated changes, suggested by the Reviewers have been introduced to the manuscript (highlighted within the document with different color).

Comment:

I suggest acceptance for publishing only some language editing.

Response:

We thank the reviewer and we carefully revised the language editing.

Sincerely

Reviewer 2 Report

The authors study aimed to evaluate selected monoterpenes in binary combination using a mixture design method against adult females of Culex pipiens. However, the data obtained are only preliminary data offering no new insigths in mechanism of their action such is the inhibition of  cholinesterases  (AChE and/or BuChE) or their SAR activity (Current Medicinal Chemistry, 2020, 27, 4297-4343).

Author Response

Dear Editor,

The authors would like to express their gratitude to the reviewer for the revision of our manuscript entitled: “Evaluation of the synergistic effect of pure molecules on adults of Culex pipiens (Diptera: Culicidae)

Comment:  

The authors' study aimed to evaluate selected monoterpenes in binary combination using a mixture design method against adult females of Culex pipiens. However, the data obtained are only preliminary data offering no new insigths in mechanism of their action such is the inhibition of  cholinesterases  (AChE and/or BuChE) or their SAR activity (Current Medicinal Chemistry, 2020, 27, 4297-4343).

Response:

The authors thank the reviewer for this remark. Effectively, this research serves as preliminary data and preliminary results for ulterior works (As perspectives). Indeed, many reports highlighted various mechanisms of action of the essential oils and their bioactive molecules including your suggested reference (Ellman et al., 1961; Ryan and Byrne, 1988; Enan, 2001; Rattan, 2010; Burčul et al., 2020). Actually, we stated some mechanisms of action in the discussion part (Lines 332-347). However, understanding the real mechanism of action of the EOs, in general, is still complex since these secondary metabolites constitute a complex mixtures of compounds, major and minor ones, and the insecticidal effect could be attributed to the major components, or minor ones, or probably the synergistic or antagonistic effect between various EOs molecules, with no individual compound making a dominating contribution (Isman et al., 2011). Similarly, it’s very difficult to detect the direct action of the pure molecules in combinations as each molecule has a specific action mode and can act together through several mechanisms (the inhibition of acetylcholinesterase, blockage of GABA-gated chloride channel, disruption and inhibition of cellular respiration, the blockage of octopamine receptors…). Indeed, the monoterpenes' action modes depending on the target sites (Choi et al. 2006). Nevertheless, further investigation is required to understand the action mode of the tested compounds (alone and in combinations), this why we think to extend the study and especially evaluate the cholinesterase inhibitory activity and we indicated this in the revised conclusion (Lines 372-374). All added information were highlighted with yellow color. 

References

Ellman, G.L., Courtney, K.D., Andres, V., Featherstone, R.M. A new and rapid color- imetric determination of acetylcholinesterase activity. Biochem. Pharmacol. 1961, 7, 88–95.

Enan, E.E. Insecticidal activity of essential oils: octopaminergic sites of action. Comp. Biochem. Physiol. 2001, 130, 325–337.

Rattan RS. Mechanism of action of insecticidal secondary metabolites of plant origin. Crop Protec. 2010, 29, 913-20.

Burčul, F., Blažević, I., Radan, M., Politeo, O. Terpenes, Phenylpropanoids, Sulfur and Other Essential Oil Constituents as Inhibitors of Cholinesterases. Current Medicinal Chemistry, 2020, 27, 4297-4343.

Isman, M.B., Miresmailli, S., MacHial, C. Commercial opportunities for pesticides based on plant essential oils in agriculture, industry and consumer products. Phytochem. Rev. 2011, 10, 197–204.

Choi, WS, Park BS, Lee YH, Jang DY, Yoon HY, Lee SH. Fumigant toxicities of essential oils and monoterpenes against Lycoriella mali adults. J. Crop Prot 2006. 25. 398–401.

Sincerely

Author Response

Dear editor,

The authors would like to express their gratitude to the reviewer for the revision of our manuscript entitled: “Evaluation of the synergistic effect of pure molecules on adults of Culex pipiens (Diptera: Culicidae)

Comment:

The idea of using essential oil components in mosquito management programs is compelling, particularly if these components are less harmful to humans, animals, and the environment than larvicides and adulticides currently used (though, importantly, the authors have not demonstrated this point).

Response:

We thank the reviewer for this comment. Indeed, EOs and their bioactives compounds constitute, as low-risk insecticides, the best alternative to the synthetic products. Thus, we demonstrated this point in the discussion part (Lines 350-352). Moreover, the authors previously checked the information about the risk of the tested compounds; they actually constitute the major constituents in Mentha pulegium and Rosmarinus officinalis EOs. Mint and Rosemary are among the herbs that are exempt from registration with the U.S. Environmental Protection Agency (EPA) under FIFRA section 25(b) regulations according to the report of Brian P. Baker and Jennifer A. Grant “Active Ingredients Eligible for Minimum Risk Pesticide Use: Overview of the Profiles”. This report listed the products that are so well established as safe and they don’t have to be registered. This helps the user to know that the product is made with ingredients that are safe and don’t present a danger to health. 

“Discussion part (Lines 350-352)”

“Indeed, EOs and their components constitute a safer and ecofriendly mosquito control tools. Many bioactive molecules are selective and less toxic to humans, animals, and the environment [43, 3, 44]”.

Main Points:

Comment 1:

There are many more terpenes than those studied here. The authors should describe why they chose to focus on these four terpenes in this study.

 In addition, they should provide a rationale for why they chose to study the pairwise combinations they did, rather than other possible pairwise combinations.

Answer 1:

We thank the reviewer for these comments. Actually, this study constitutes a continuity of other investigations that are planned to be achieved in the context of the thesis; it constitutes the second part of our paper that has been accepted recently to be published in plants journal “Ramzi, A.; El Ouali Lalami, A.; Ez zoubi, Y.; Assouguem, A.; Almeer, R.; Najda, A.; Ullah, R.; Ercisli, S.; Farah, A. Insecticidal Effect of Wild-Grown Mentha pulegium and Rosmarinus officinalis Essential Oils and Their Main Monoterpenes against Culex pipiens (Diptera: Culicidae). Plants 2022, 11(9), 1193; https://doi.org/10.3390/plants11091193”. The four terpenes tested in our study through the binary mixtures constitute the main compounds in Mentha pulegium and Rosmarinus officinalis essential oils. We evaluated both essential oils and their major compounds alone in the above-mentioned paper. Moreover, we used only the four monoterpenes in our research since they were the available ones during the pandemic of Covid-19, and we couldn’t extend the insecticidal activity using other monoterpenes because of the pandemic circumstances noting the purchase of the molecules, the access to the laboratory, the collect of the larvae from the breeding sites… However, we plan to test others monoterpenes and also sesquiterpenes alone and in combinations, we indicated this as perspectives in the conclusion part (374-376).

For the use of the pairwise combinations, we would like to inform the Reviewer that our main and first objective was to test all possible binary combinations and also ternary combinations. However, we faced some difficulties to collect mosquitoes during the pandemic. Moreover, in the post-pandemic, the significant fluctuation of temperature here in Morocco (high-temperature level during the day and a cold climate at night), especially during the two last years, has had a negative impact on the mosquito cycle life, so we couldn’t collect the appropriate density of larvae for testing all possible combinations. Besides, the process from collect the larvae and carrying out the test takes a long time. Thus, all of these obstacles prevented us to achieve the first goal. However, evaluate the other possible mixtures still scheduled in our thesis goals and of course, taking into account compatible mixtures of compounds that should be used in a bio-insecticide formulation, so we plan to extend the research by providing a clear and consistent finding on the effect of the EOs bioactive molecules and we indicated this information in the conclusion part (as perspectives; lines374-376).

Comment 2:

The authors should provide more detail about how the mixtures were prepared. Is the “component proportion” referring to volume, mass, moles, or something else? 

Answer 2:

The authors provided more detail about how the mixtures were prepared (paragraph 2.2. Preparation of the binary mixture; Lines 95-99). The “component proportion” is referring to volume; we indicated this in the above-mentioned paragraph.

Paragraph 2.2. Preparation of the binary mixture; Lines 95-99:

“Based on the LC50 values (Table 1) obtained for the individual compounds in a previous study (Ramzi et al., 2022) [22], seven binary combinations for 1,8 cineole+α-pinene and carvone+ R(+)-pulegone were prepared using binary mixture design as described in the Table 2. For each experiment, the two compounds were combined using the various proportions that are referring to volume.”

Comment 3:

A negative control needs to be performed and reported.

Answer 3:

Actually, we used a negative control; Dimethylsulfoxyd (DMSO), we just forgot to mention it in the document. We indicated this in the paragraph 2.4. Fumigant toxicity (Line 132) with green color.

Comment 4:

The authors should provide a justification for the trend lines in Figure 3. In the current presentation, there are minima and maxima that are not supported by experimental data: are these maxima and minima real? What is the mathematical model that supports the fitting of the data in this way?

Answer 4:

We thank the reviewer for noticing this inconsistence and we would like to apologize for these mistakes, it was just a typing error, thus, the data presented in Table 3 has been corrected. For Figure 3, the justification for the trend lines has been added to the results section, the maximum response zones (Isoresponses) part, lines 193-196. The adopted mathematical model for two mixtures has been added in the manuscript in the result section, statistical validation of the selected model part, lines 181-190. All corrections have been highlighted in the document with Green color.

Additional Points

Comment 1:

The superscript formatting in Table 1 needs to be fixed.

Answer 1:

We carefully revised the Table 1.

Comment 2:

In Table 2, periods should be used in place of commas in all numbers and the number of reported digits should be consistent (two digits is probably appropriate).

Answer 2:

We corrected the Table 2

Comment 3:

All the figures are difficult to read because the labels are small and blurry. Figures need to be remade to be more legible.

Answer 3:

Authors remade all figures, they are more legible.

Comment 4:

Citation formatting needs to be made consistent.

Answer 4:

We carefully revised the citation formatting (indicated in the manuscript with green color).

Comment 5:

A reference should not be used as the subject of a sentence, as it is on page 9, line 252.

Answer 5:

We corrected this in the sentence indicating the citation (Line 334), and we corrected all citation mistakes.  We also revised some missing citations and the references part (all revision were indicated with green color).   

Comment 6:

The abbreviation “EO” should be spelled out before it is used.

Comment 6:

The abbreviation “EO” was indicated in the introduction part (Line 75).

Sincerely

Round 2

Reviewer 2 Report

The authors improved the discussion part using appropriate references. However, the conclusions should be based on more concrete experiments that explain the action of essential oil components.

Reviewer 3 Report

The revised document addressed sufficiently what I would call the "cosmetic" concerns that I raised (clarity of figures, use of significant figures) and some of the scientific questions as well.

However, I am very concerned to see that the crucial data in the paper (Table 3) have been completely altered, because of a typo (according to the authors) but that none of the other figures or tables have been changed or corrected. Unless the authors can convince me that the (completely different) data in Table 3 do not have any effect on the later tables or figures in the paper, I cannot support this paper for publication. More seriously, the authors are displaying a concerning lack of attention to detail in this submission.